# The Impact of PCSK9 on Diabetic Cardiomyopathy: Mechanisms and Implications

**DOI:** 10.3390/biom15091240

**Published:** 2025-08-27

**Authors:** Haixia Wang, Pei Wang, Yubo Wang, Shuzhen Du, Jing Zhao, Zheng Zhang

**Affiliations:** 1The First School of Clinical Medicine, Lanzhou University, Lanzhou 730000, China; wanghx2024@lzu.edu.cn (H.W.); pwang20@lzu.edu.cn (P.W.); wyubo2024@lzu.edu.cn (Y.W.); dushuzh2023@lzu.edu.cn (S.D.); 2Department of Heart Center, The First Hospital of Lanzhou University, Lanzhou 730000, China

**Keywords:** PCSK9, DCM, PCSK9 inhibitor, inflammatory response, NLRP3 inflammasome, lipid metabolism

## Abstract

Diabetic cardiomyopathy (DCM) is a common and clinically relevant complication of diabetes mellitus, defined by myocardial dysfunction in the absence of overt coronary artery disease or systemic hypertension. Recent studies have identified proprotein convertase subtilisin/kexin type 9 (PCSK9) as a pivotal mediator in the pathogenesis of DCM. PCSK9 contributes not only to dyslipidemia via degradation of LDLR and consequent elevation of circulating LDL-C, but also to metabolic derangements and inflammation through interactions with receptors such as CD36 and Toll-like receptor 4 (TLR4). In DCM, PCSK9 has been shown to exacerbate inflammation and pyroptosis and is closely linked to impaired autophagic function. Elevated circulating PCSK9 has emerged as a potential biomarker for cardiovascular events in patients with type 2 diabetes mellitus (T2DM). At the same time, long-term administration of PCSK9 inhibitors (PCSK9i) has not been associated with a significant increase in incident diabetes. Furthermore, PCSK9 loss-of-function mutations have been linked to a modestly heightened risk of T2DM, underscoring its complex involvement in cardiometabolic regulation and disease. This review synthesizes current insights into the mechanistic and therapeutic roles of PCSK9 in DCM, aiming to inform precision cardiovascular risk management strategies in T2DM populations.

## 1. Introduction

Diabetes mellitus is a chronic metabolic disease that poses a serious threat to human health. Currently, type 2 diabetes mellitus (T2DM) affects approximately 540 million individuals worldwide, including 6.1 million in Europe. Projections estimate that by 2045, the global prevalence of diabetes will rise to 783 million, representing 12.2% of the worldwide population [1,2]. T2DM is the most prevalent form of diabetes, with cardiovascular complications constituting one of the leading causes of mortality [3]. Diabetic cardiomyopathy (DCM) is defined as myocardial systolic and/or diastolic dysfunction in the presence of diabetes mellitus. It is not dependent on coronary artery disease, hypertension, or other cardiovascular risk factors [2].

The pathogenesis of DCM is complex and involves multiple pathological processes, such as inflammatory response, oxidative stress, cardiomyocyte death, autophagy, metabolic disorders, insulin resistance, mitochondrial damage, endoplasmic reticulum stress, and accumulation of advanced glycation end products (AGEs) [4,5,6,7]. In recent years, proprotein convertase subtilisin/kexin type 9 (PCSK9), a key regulator of lipid metabolism, has been increasingly recognised for its non-canonical roles in glucose metabolism, inflammation, and cell death. Notably, PCSK9 expression is markedly elevated in T2DM and has been shown to correlate positively with major adverse cardiovascular events (MACEs), suggesting its potential pathogenic involvement in DCM warrants further investigation.

PCSK9 plays a pivotal role in lipid metabolism by regulating low-density lipoprotein receptor (LDLR) degradation. Beyond this canonical function, PCSK9 influences fatty acid uptake, inflammatory pathways, and mitochondrial function through its interactions with key signalling mediators such as CD36 and Toll-like receptor 4 (TLR4), exerting profound effects on myocardial homeostasis [8]. In addition, PCSK9 is involved in the pathogenesis of various diseases such as atherosclerotic cardiovascular disease, ischemia/reperfusion injury, cancer, and HIV [9,10].

Although Mendelian randomisation studies [11], meta-analyses [12], and clinical studies [13,14] suggest that PCSK9 inhibitors (PCSK9i) may slightly affect glucose metabolism, large-scale clinical studies have not found a significant increase in the risk of new-onset diabetes mellitus (NODM) [15,16] (Table 1). This suggests an apparent cardiovascular protective effect of PCSK9i. Although the conclusions of related studies have shown significant discrepancies, the role of PCSK9 and its inhibitors in diabetes and the effects of glucose-lowering drugs on PCSK9 have attracted much attention in recent years. In recent years, the interaction between PCSK9 and glucose-lowering drugs, such as the modulation of PCSK9 expression by SGLT2 inhibitors and GLP-1 receptor agonists, has also triggered research interest.

This review aims to systematically summarise the pathological mechanisms and recent advances concerning PCSK9 in DCM, focusing particularly on its cell-type-specific functions within the heart. It further explores the cross-talk between PCSK9 and key regulatory pathways, including lipid and glucose metabolism, inflammation, autophagy, and ferroptosis, while highlighting its potential clinical value as a therapeutic target.

## 2. Mechanisms of Diabetic Cardiomyopathy

DCM is characterised by myocardial hypertrophy and progressive impairment of cardiac function, which may ultimately culminate in heart failure [17]. Individuals with T2DM have approximately a 2.4-fold higher risk of developing heart failure compared to non-diabetic individuals [18]. More than 25% of patients with T2DM exhibit subclinical structural or functional cardiac abnormalities, with left ventricular diastolic dysfunction being the most prevalent. As the disease progresses, diabetic patients may develop heart failure with preserved ejection fraction (HFpEF) or reduced ejection fraction (HFrEF), with a particularly significant risk in female patients [2,19].

Metabolic dysregulation is a central pathological hallmark of DCM. Recent studies have identified the TGR5 signalling pathway as a critical regulator of lipid metabolism and myocardial function. Loss of TGR5 promotes aberrant membrane trafficking of CD36 through DHHC4-mediated palmitoylation, leading to intracellular lipid accumulation and compromised cardiac performance. Conversely, TGR5 activation significantly limits abnormal fatty acid influx into cardiomyocytes, exerting cardioprotective effects [20,21]. This novel mechanism expands the understanding of myocardial lipotoxicity and highlights the “TGR5–CD36” axis as a promising therapeutic target for DCM.

Diabetes is increasingly recognised as a chronic, low-grade inflammatory disease. Experimental studies have demonstrated that inhibition of reactive oxygen species (ROS) production significantly attenuates myocardial pro-inflammatory signalling, underscoring the ROS–NLRP3 inflammasome axis as a key molecular bridge linking metabolic dysregulation to cardiac injury [22,23]. In diabetic conditions, high-fat-induced mitochondrial ROS generation leads to mitochondrial damage and mitochondrial DNA (mtDNA) leakage, which activates the cytosolic DNA sensor cyclic GMP–AMP synthase (cGAS). It promotes stimulator of interferon genes (STING) translocation to the Golgi apparatus, subsequently activating IRF3, NF-κB, and thioredoxin-interacting protein (TXNIP), culminating in inflammation, apoptosis, and myocardial fibrosis [24,25,26]. Concurrently, increased ROS in the diabetic atrial myocardium stimulates the NF-κB pathway, which directly binds the Atrogin-1 promoter and enhances FBXO32 gene transcription, thereby accelerating SK2 protein degradation in an Atrogin-1-dependent manner [27].

AGEs further exacerbate inflammation by forming a ternary complex with TLR4 and MD2, activating pro-inflammatory cascades in the diabetic milieu [28]. Artesunate has been shown to target MD2 and disrupt this pathway, thereby reducing fibroblast proliferation, migration, and collagen deposition, ultimately improving cardiac function [29]. AGEs can also promote NLRP3 inflammasome activation via ROS, triggering inflammation and pyroptosis and impairing tissue repair in diabetic settings [30].

Chronic low-grade inflammation is a key player in driving insulin resistance (IR) in the context of obesity and T2DM. Inflammatory cascades begin with IgG deposition in adipose tissue, which triggers macrophage infiltration and disrupts insulin signalling [31]. Hyperactivation of CREBZF exacerbates adipose tissue inflammation and IR by competitively binding NF-κB p65 with inhibitor of NF-κB alpha (IκBα) [32]. Additionally, hypoxia-inducible factor 2α (HIF-2α) in macrophages maintains immunometabolic homeostasis through the H3K27me3–Cpt1a axis and suppresses activation of the NLRP3 inflammasome [33]. JJMJD8 enhances LPS-induced inflammation and worsens IR by interacting with interferon regulatory factor 3 [34] (Figure 1).

Branched-chain amino acids promote polarisation of macrophages toward the pro-inflammatory M1 phenotype via activation of the Janus kinase 1/signal transducer and activator of transcription 1 (JAK1/STAT1) pathway, thereby impairing insulin sensitivity [35]. Activation of c-Jun N-terminal kinase (JNK) and p38 mitogen-activated protein kinases is also strongly linked to IR [36] (Figure 1). Lactate metabolism also modulates immune responses. MCT1 facilitates lactate uptake into mitochondria, supporting M2 anti-inflammatory polarisation, while MCT4 promotes lactate export and M1 polarisation. In pancreatic β-cells, lactate suppresses insulin release by activating GPR81 and inhibiting the cAMP–PKA pathway [37].

SOSTDC1, secreted by CD4^+^ T cells, is elevated in obese individuals and mice on a high-fat diet. It promotes fat accumulation and worsens IR by boosting adipogenesis, blocking lipolysis, and activating the lipoprotein receptor-related protein 5 (LRP5)/6 β-catenin pathway [38]. Furthermore, endothelial loss or pharmacologic blockade of the adrenomedullin receptor ameliorates obesity-induced IR [39]. Statin therapy has also been implicated in worsening IR by reducing circulating glucagon-like peptide-1 (GLP-1) levels in a gut microbiota-dependent manner [40].

## 3. Mechanisms of PCSK9

### 3.1. PCSK9 Biology

PCSK9, a member of the proprotein convertase family, plays a pivotal role in the proteolytic activation, modification, and degradation of secreted proteins. The protein comprises 692 amino acids, with a molecular mass of approximately 74.3 kDa, and contains three characteristic functional domains arranged from the N-terminus: a prodomain, a catalytic domain, and a cysteine–histidine-rich C-terminal domain (CRD) [41,42,43]. PCSK9 is expressed primarily in the liver, kidneys, and intestines, but is also detectable in other tissues, including the lungs, pancreas, skin, and cerebrospinal fluid [44]. Notably, PCSK9 expression has also been identified in cardiomyocytes, endothelial cells (ECs) [45], macrophages [46], monocytes, and vascular smooth muscle cells (VSMCs) [47,48]

PCSK9 modulates several membrane-bound receptors on pancreatic β-cells, including very low-density lipoprotein receptor (VLDLR), CD36, and fatty acid transport proteins, thereby influencing insulin secretion and fatty acid uptake, and ultimately contributing to glucose homeostasis [49]. Circulating PCSK9 targets monoclonal antibodies that spatially site-block PCSK9 binding to the EGF-A structural domain of LDLR in hepatocytes and other tissues by recognising its catalytic subunit [50]. Interestingly, PCSK9-mediated regulation of VLDLR is independent of LDLR, as it forms an internalisation complex that facilitates lysosomal degradation through selective EGF-A domain binding [51]. Emerging evidence suggests that PCSK9 may further accelerate LDLR degradation by interfering with SNX17-dependent recycling mechanisms [52]. Beyond its receptor-level effects, PCSK9i has shown significant clinical efficacy in lowering LDL-C levels (*p* < 0.01), improving atherogenic lipid indices, including TC, non-HDL-C, TG, Lp(a), and ApoB (all *p* < 0.05), while simultaneously elevating HDL-C concentrations (*p* < 0.05) [53,54].

### 3.2. PCSK9 Demographics

PCSK9i, such as evolocumab, are suitable for patients across all age groups. Notably, initiating long-term evolocumab therapy in older adults with atherosclerotic cardiovascular disease yields cardiovascular benefits comparable to, if not exceeding, those seen in younger individuals [55]. Recent analyses further confirm that the efficacy and safety profile of evolocumab remains consistent in patients aged ≥75 years compared to those under 75 [56].

Emerging genomic data suggest that the metabolic effects of PCSK9 inhibition may vary by ancestry. Rosoff et al. [57] reported that prolonged pharmacologic or genetic inhibition of PCSK9 exerts a broadly neutral impact on glycemic control and T2DM risk among most non-European populations. However, a slightly increased risk was noted in African ancestry cohorts. Conversely, inhibition of HMGCR, the molecular target of statins, was linked to a higher incidence of T2DM in individuals of South Asian, East Asian, and European descent. Importantly, PCSK9-related genetic modulation did not significantly alter hepatic enzyme levels—including ALT, AST, GGT, and ALP—across all ethnic groups examined, supporting a reassuring hepatic safety profile. Moreover, variants associated with reduced PCSK9 expression and lower circulating PCSK9 protein were correlated with increased direct bilirubin levels, a finding in line with observational evidence indicating an inverse relationship between PCSK9 and bilirubin concentrations [58].

Sex has been identified as a biological determinant of plasma PCSK9 concentrations in patients with diabetes, with variations observed across different age strata. PCSK9i significantly reduces LDL-C and MACE risk in both sexes, with no significant sex differences in MACE reduction. However, LDL-C reduction is greater in males [59]. Women tend to initiate monoclonal antibody therapy targeting PCSK9 at a later age than men. In females, circulating PCSK9 concentrations increase with age, particularly following menopause, which may reflect enhanced therapeutic responsiveness in older versus younger women. Furthermore, sex—rather than age—was a stronger determinant of baseline LDL-C levels and the likelihood of achieving treatment goals [60].

### 3.3. PCSK9 and Lipid Metabolism

#### 3.3.1. HNF1α

HNF1α serves as a central transcriptional activator in hepatic metabolic regulation by binding its conserved recognition sequence (5′-GTTAATNATTAAC-3′) via the N-terminal POU domain, thereby directly promoting PCSK9 transcription [61]. Conversely, FOXO3 functions as a negative regulator by competitively occupying the PCSK9 promoter, thereby attenuating HNF1α transcriptional activity [62]. Insulin negatively regulates HNF1α via the mTORC1–PKCδ signalling axis, decreasing PCSK9 expression and elevated LDLR levels [63]. The small-molecule compound DC371739 also downregulates PCSK9 by specifically inhibiting HNF1α-mediated transcriptional activity [64]. In terms of genetic regulation, Liu et al. [65] demonstrated that the MODY3-associated HNF1A IVS7-6G>A mutation induces aberrant mRNA splicing and yields truncated proteins lacking the transactivation domain, with the 282–501 region shown to be critical for HNF1A function. In addition, Jorge et al. [66] found that the long non-coding RNA HASTER modulates hepatic and pancreatic β-cell metabolic function by regulating HNF1A expression. Recent mechanistic evidence suggests that the HNF1α–PCSK9–LDLR signalling cascade may be involved in the lipid-modulatory effects of dapagliflozin [67] (Appendix A).

#### 3.3.2. SREBP2

Sterol regulatory element-binding protein 2 (SREBP2) plays a pivotal role in regulating intracellular cholesterol homeostasis by modulating genes involved in lipid metabolism. Through its basic helix-loop-helix structural motif, SREBP2 selectively binds sterol regulatory elements (SREs) within the promoter regions of target genes, including PCSK9, LDLR, and HMGCR, thereby initiating their transcription [68] (Appendix A). In addition, Austin et al. [69] reported that caffeine inhibits SREBP2 transcriptional activity by elevating endoplasmic reticulum calcium levels in hepatocytes, which leads to increased LDLR expression and improved LDL-C clearance.

SREBP2 activation is also modulated by inflammatory signalling. Proinflammatory cytokines such as TNF-α and IFN-γ upregulate PCSK9 expression in hepatocytes through synergistic activation of SREBP2 [70]. Additionally, IL-1β enhances mTORC1 activity via the NF-κB pathway, impacting SREBP2 maturation and downstream LDLR regulation [71].

#### 3.3.3. CD36

CD36 is a pivotal fatty acid translocase that regulates transmembrane fatty acid transport via a dynamic membrane recycling mechanism. Aberrant intracellular localisation of CD36 has been implicated in multiple metabolic disorders [72,73,74]. In diabetes, persistent surface localisation of CD36 promotes excessive fatty acid uptake, contributing to lipid imbalance. This process is modulated through a stable, non-degradable complex formed with PCSK9, positioning CD36 as a regulatory node within the gut–adipose tissue lipid axis [75,76]. Furthermore, exogenous PCSK9 has been shown to reduce basal fatty acid uptake in cardiomyocytes independently of CD36 expression, suggesting alternative regulatory pathways [77]. In peripheral nerves, PCSK9 deficiency results in CD36 upregulation, mitochondrial structural damage, lipid accumulation, and elevated acylcarnitine levels, ultimately leading to peripheral neuropathic symptoms [78]. CD36 modulates SREBP1 processing by interacting with insulin-induced gene 2 (INSIG2), thereby maintaining hepatic lipid homeostasis [79] (Appendix A).

### 3.4. PCSK9 and Glucose Metabolism

#### 3.4.1. PCSK9 and T2DM

The functional role of PCSK9 within pancreatic islet cells remains controversial. Although circulating and liver-derived PCSK9 appear to exert minimal influence on pancreatic β-cell physiology, targeted PCSK9 deficiency within β-cells has been associated with intracellular cholesterol accumulation and impaired insulin secretion, suggesting a context-specific requirement for endogenous PCSK9 in maintaining islet homeostasis [80]. Conversely, other preclinical studies have shown that extensive inactivation of PCSK9 in β-cells does not alter glycemic control or insulin output, raising questions about its functional necessity in this compartment [8,15]. Emerging evidence indicates that δ-cell-secreted PCSK9 may play a paracrine protective role by limiting cholesterol uptake in β-cells, thereby attenuating lipotoxicity under metabolic stress conditions [81]. Insulin itself induces PCSK9 expression via the ERK signalling cascade in a dose- and time-dependent manner, forming a potential autocrine feedback loop [44].

From a clinical perspective, elevated serum PCSK9 concentrations have been documented in individuals with impaired glucose metabolism, including those with impaired glucose tolerance and T2DM [82]. In pregnant women, PCSK9 levels were significantly higher in the gestational diabetes mellitus (GDM) group than those with standard glucose tolerance (NGT), suggesting a possible pathophysiological role for PCSK9 in GDM-related dyslipidemia and glycemic dysregulation. Furthermore, Wu et al. [83] reported that serum PCSK9 levels were positively correlated with LDL-C, fasting plasma glucose, and HbA1c in GDM patients, reinforcing its potential role as a metabolic biomarker. These findings suggest that PCSK9’s role in islet biology may depend highly on cellular context and metabolic milieu. Further, mechanistic and translational studies are needed to elucidate its contribution to β-cell homeostasis.

Current evidence suggests that prolonged use of PCSK9i does not confer a heightened risk of developing NODM [84,85]. Among prediabetic individuals, no significant difference in NODM incidence was observed between treatment and control arms, and glycemic indices in patients with established T2DM remained unaffected by PCSK9i exposure [14,80]. Despite these findings, the long-term influence of PCSK9i on glucose metabolism remains an area of active investigation, particularly in light of the need to evaluate subgroup-specific susceptibility and the durability of glycemic neutrality. Given these uncertainties, further studies are warranted to examine the metabolic interplay between PCSK9 signalling and hypoglycemic therapies.

PCSK9i has consistently demonstrated substantial cardiovascular benefits in patients with diabetes mellitus. Meta-analyses and large-scale randomised controlled trials report an approximate 18% relative risk reduction in MACEs among diabetic patients receiving PCSK9i therapy, alongside significant improvements in lipid indices [86,87]. The FOURIER and FOURIER-OLE studies provided evidence of evolocumab’s efficacy, showing significant long-term MACE reduction in both multivessel disease (MVD) and non-MVD subgroups. While the MVD cohort derived earlier and more robust benefits, patients without MVD also exhibited a sustained 25% risk reduction over time, underscoring the broad clinical utility of PCSK9i across varying cardiovascular risk strata [88].

**Table 1 biomolecules-15-01240-t001:** The Relationship Between PCSK9i, Lipid Profiles, and NODM (The arrows in the table indicate the changes: downward arrows indicate a decrease, while hyphens (–) represent no significant effect observed).

Author	Dose of the Intervention	Year	Number of Participants	Follow-Up (Mean or Median) Years	Inclusion Criteria	Type Randomization	Outcome Change
Imbalzano et al. [89]	Alirocumab/ Evolocumab Varied (see individual trials)	2023	20651(aggregated)	51 weeks	DM + hypercholesterolemia, RCTs comparing PCSK9i vs. placebo	Meta-analysis of 8 RCTs	MACE 18% ↓; LDL-C, HDL-C, TG, Lp(a), and ApoB ↓.
Ray et al. [90]	Alirocumab 75 mg every 2 weeks (Q2W), increased to 150 mg if required	2019	413	24 weeks	Type 2 Diabetes Mellitus (T2DM), mixed dyslipidemia, LDL-C ≥ 100 mg/dL, ASCVD, on stable maximally tolerated statin therapy	RCT	LDL-C, non-HDL-C, ApoB, and LDL particle number (LDL-PN) ↓
Fischer et al. [91]	Alirocumab (75 mg or 150 mg) and Evolocumab (140 mg)	2021	237	Median 18 months	Age > 18 years; PCSK9 inhibitors (alirocumab or evolocumab) for secondary prevention; Available LDL-C data at baseline and during follow-up for at least 3 months	Observational study	MACE ↓; LDL-C, non-HDL-C, and ApoB ↓
Rosenson et al. [92]	Evolocumab 420 mg subcutaneously once a month	2019	421	12 weeks	Type 2 diabetes, hyperlipidemia or mixed dyslipidemia, background atorvastatin 20 mg/d	RCT	LDL-C 54.3% ↓ (12 weeks); non-HDL-C, and ApoB ↓.
Chen et al. [93]	Evolocumab 140 mg every 2 weeks/420 mg monthly	2019	453	12 weeks	T2DM with hyperlipidemia, LDL-C ≥ 2.6 mmol/L on statin or ≥3.4 mmol/L without statin	RCT	LDL-C ↓ Non-HDL-C, ApoB100, triglycerides, and Lp(a) ↓; HbA1c and FSG(–)
Schwartz et al. [16]	Alirocumab (75 mg or 150 mg every 2 weeks)	2025	8107	Median follow-up of 2.4 years	Patients with recent acute coronary syndrome, elevated lipoproteins, and no diabetes at baseline	RCT	NODM(–)
Sabatine et al. [94]	Evolocumab 140 mg every 2 weeks or 420 mg monthly	2017	27,564	104 weeks	Atherosclerotic cardiovascular disease, LDL-C ≥ 1.8 mmol/L, on statin therapy	RCT	NODM(–)
Moura et al. [95]	Evolocumab 140 mg every 2 weeks vs. placebo	2025	9388 (No diabetes at baseline)	Median follow-up of 2.3 years	Age 40–85, stable cardiovascular disease, LDL ≥ 70 mg/dL, on statin therapy, no T2D at baseline	RCT	NODM(–)
González-Lleó et al. [14]	Alirocumab 75 or 150 mg or evolocumab 140 mg every 2 weeks	2024	218	Mean follow-up of 3.2 years	Patients over 18 years with hypercholesterolemia, including familial hypercholesterolemia (FH), undergoing treatment with PCSK9 inhibitors	Observational study	NODM (2.6%/year); Glycemic parameters(–)

#### 3.4.2. PCSK9 and Antihyperglycemic Drug

Emerging evidence indicates a complex and potentially synergistic relationship between PCSK9 signalling and antidiabetic pharmacotherapies. The current therapeutic landscape for DCM is dominated by SGLT-2 inhibitors, GLP-1 receptor agonists, and their combined use. While efficacious in glycemic control, these agents often induce gastrointestinal adverse effects that compromise long-term adherence. Combination regimens have shown enhanced clinical outcomes, including reductions in HbA1c, fasting plasma glucose, and body weight compared to monotherapy [96]. On a mechanistic level, liraglutide was found to suppress hepatic PCSK9 and LDLR expression through an HNF1α-dependent pathway in both HepG2 cells and db/db mice [97], and also downregulate PCSK9 transcription in adipose and hepatic tissues while promoting lipoprotein lipase expression in high-fat diet–induced mouse models [98].

Dapagliflozin similarly modulates hepatic lipid metabolism through the PCSK9/LDLR pathway [67]. A study showed no potentially deleterious effects on circulating PCSK9 levels in 78 patients with T2DM treated with 25 mg/d of empagliflozin for 4 weeks [99]. In animal models, dapagliflozin suppressed PCSK9 expression by inhibiting carbohydrate-responsive element-binding protein activity, resulting in reduced hepatic and serum PCSK9 levels in both diabetic and nondiabetic dyslipidemic mice [100]. Furthermore, PCSK9i and atorvastatin reduce cardiac impairment in ovariectomized prediabetic rats via improved mitochondrial function and Ca2+ regulation [101]. However, not all outcomes were favorable: alirocumab worsened renal dysfunction in obese ZSF1 rats, and empagliflozin failed to prevent kidney disease progression in the same model, despite its renoprotective effects in human diabetic kidney disease [102].

Insulin exerts a pivotal influence on both systemic and hepatic expression of PCSK9. Miao et al. [103] reported that insulin receptor knockdown in murine hepatocytes led to significant reductions in both PCSK9 mRNA and protein levels, highlighting the dependence of PCSK9 expression on insulin signalling. In parallel, glucagon was shown to suppress the PCSK9 gene and protein expression by approximately 50% in primary rat hepatocytes. Niesen et al. [104] consistently observed a marked decrease in hepatic PCSK9 protein levels in streptozotocin-induced insulin-deficient Sprague-Dawley rats. These hormonal effects appear to be mediated through post-transcriptional pathways, with PCSK9 acting as a downstream effector in the glucagon- and estrogen-induced regulation of LDL receptor availability [105].

#### 3.4.3. PCSK9 and Atherogenic Dyslipidemia

Circulating PCSK9 has emerged as a potential prognostic biomarker for cardiovascular (CV) events in patients with T2DM [45,106,107]. While glucose metabolism alone does not appear to influence plasma PCSK9 concentrations directly, T2DM may modulate the relationship between PCSK9 and atherogenic lipid markers such as non-high-density lipoprotein cholesterol (non-HDL-C) and apolipoprotein B (ApoB) [108]. Several studies have demonstrated a significant positive correlation between circulating PCSK9 and LDL-C levels [109,110]. One cohort study found an inverse association between genetically determined LDL-C levels and T2DM risk, suggesting that PCSK9-mediated lipid regulation may have complex metabolic implications [111].

In a cohort of 2984 participants, higher serum PCSK9 levels were significantly associated with multivessel coronary heart disease (CHD) and elevated Gensini scores. They were identified as an independent predictor of CHD and MACEs in individuals with T2DM [107]. Furthermore, PCSK9 levels were found to predict carotid–femoral pulse wave velocity, accounting for 16.9% of its variance, thereby supporting its utility in cardiovascular risk stratification for diabetic patients [112]. Importantly, PCSK9i appear to reduce MACE risk and improve lipid profiles in patients with both diabetes and dyslipidemia, irrespective of baseline glycemic status [89] (Table 1).

#### 3.4.4. PCSK9 Mutations and Gene Silencing

Genetic polymorphisms affecting PCSK9 function are increasingly recognised for their metabolic consequences. Several PCSK9 loss-of-function (LOF) variants, including rs11583680, rs11591147, rs2479409, and rs11206510, have been associated with modest elevations in fasting glucose, increased body weight, and a higher risk for T2DM development [113]. In contrast, gain-of-function (GOF) mutations such as S127R and R496W reduce PCSK9’s affinity for LDL particles, augmenting LDLR degradation and exacerbating hypercholesterolemia [114]. Paradoxically, LOF mutations in LDLR observed in patients with familial hypercholesterolemia appear to reduce T2DM risk, suggesting gene-specific metabolic compensation. Mechanistic studies in Pcsk9 knockout mice have demonstrated impaired mitochondrial bioenergetics in cardiac tissue, with diminished electron transport chain (ETC) activity and a shift toward glycolytic reliance that fails to sustain adequate energy output under physiological stress [115].

Therapeutically, gene silencing of PCSK9 is being explored as a novel strategy for mitigating DCM. Sustained suppression of PCSK9 expression has been achieved through the transient delivery of ethyl transfer RNA (ETR), leading to durable reductions in LDL-C [49]. In vitro studies using human microvascular endothelial cells (HMEC-1) have shown that PCSK9 silencing attenuates hyperglycemia-induced inflammatory signalling, oxidative stress, and lipid metabolic dysfunction, primarily via inhibition of the lectin-like oxidised LDL receptor-1 (LOX-1) pathway [45]. Recent advances in gene editing have further enabled epigenetic suppression of PCSK9 through targeted methylation. It is achieved by fusing catalytically inactive dCas9 to either DNA methyltransferase or demethylase domains, allowing for site-specific epigenetic modulation of the PCSK9 promoter guided by single-guide RNAs, resulting in efficient downregulation of PCSK9 and reduced LDL-C levels [116].

### 3.5. PCSK9 and Inflammation

#### 3.5.1. PCSK9 Activates the TLR4/NF-κB Pathway

PCSK9 directly interacts with TLR4 via its cysteine-rich domain (CRD), activating the TLR4/MyD88/NF-κB signalling pathway and promoting the expression of key pro-inflammatory mediators, including TNF-α, IL-1β, and tissue factor [117,118,119,120]. Gene silencing of PCSK9 has been shown to inhibit NF-κB nuclear translocation and dampen inflammatory responses [51,121]. In addition, PCSK9 enhances the activity of NLRP3 Inflammasomes by activating this pathway, creating a positive feedback loop that exacerbates myocardial fibrosis and sepsis-associated endothelial dysfunction [122,123,124]. Interestingly, the TLR4 agonist lipopolysaccharide (LPS) has been found to reverse the upregulation of PCSK9, reinforcing an inflammation-metabolism loop. Furthermore, deletion of low-density LRP5—a known transporter of PCSK9—markedly reduces PCSK9 release and downstream TLR4 activation [125].

PCSK9 activates the NLRP3 inflammasome signalling pathway by inducing damage to mtDNA [126]. It regulates Smac translocation from mitochondria to cytosol, mediates hypoxia-induced apoptosis, and inhibits neovascularisation [127]. Clinical myocardial samples have demonstrated pronounced co-localisation of PCSK9 with N-terminal gasdermin D (GSDMD-NT) in the peri-infarct zone, implicating PCSK9 in pyroptotic injury during acute myocardial infarction [128]. Mechanistic studies have demonstrated that PCSK9 activates the NLRP3 inflammasome signalling pathway by inducing mtDNA damage, ultimately triggering caspase-1-dependent pyroptosis. In PCSK9 knockout (PCSK9-/-) mice, NLRP3 signalling was inhibited, with significant reductions in GSDMD-NT expression and lactate dehydrogenase release, supporting PCSK9’s role in the regulation of inflammatory cell death [126].

#### 3.5.2. NLRP3 Inflammasome Via IL-1β Regulates PCSK9 Secretion

NLRP3 inflammasome also inversely regulates PCSK9 expression [44]. In vitro experiments had shown that ATP or Nigericin stimulation following LPS priming had led to increased PCSK9 expression in macrophages, with the effect being strictly dependent on apoptosis-associated speck-like protein containing a CARD (ASC), caspase-1, and IL-1β [44,109]. IL-1β further induces PCSK9 secretion through the MAPK signalling axis, revealing its role as a key molecular hub linking immuno-inflammation and lipid metabolism [44,126,129]. The CANTOS clinical trial validated the clinical translational value of this mechanism: the IL-1β monoclonal antibody canakinumab significantly reduced the recurrence of MACE and HF hospitalisation rates in patients with myocardial infarction, while lowering hs-CRP levels [130,131]. Furthermore, dapagliflozin suppresses PCSK9 expression through the NLRP3/IL-1β axis in a cardiotoxicity context, reinforcing the importance of this feedback network in cardiometabolic modulation [132].

### 3.6. Pyroptosis

Oxidative stress, endoplasmic reticulum stress, and immune activation induced by hyperglycemia are primary drivers of focal cellular death in diabetes. Elevated glucose concentrations stimulate the expression of multiple classes of endogenous non-coding RNAs—including miRNAs, circRNAs, and lncRNAs—which modulate transcriptional and post-transcriptional pathways involved in inflammation and programmed cell death [133].

Mechanistically, persistent hyperglycemia activates the CaMK2α/O-GlcNAcylation positive feedback loop in endothelial cells, which remains active even after restoring normoglycemia. It promotes phosphorylation of Stat1, enhances transcription of miR-15/16 precursors, increases secretion of small extracellular vesicle-associated miR-15/16, and sustains a state of chronic low-grade inflammation [134]. circGlis3 expression is significantly upregulated in both genetically and diet-induced obese mice. circGlis3 appears to exert a protective role by promoting insulin synthesis and secretion and binding to the pro-apoptotic protein SCOTIN in a caspase-3–dependent manner, thereby inhibiting β-cell apoptosis [135]. Furthermore, PCSK9 has been shown to influence mitochondrial function and apoptosis-related signalling in VSMCs, suggesting a possible contribution to diabetic vascular remodelling [126,136].

Activation of the NLRP3 inflammasome appears to be a critical pathogenic node linking programmed cell death to inflammatory amplification in diabetic vasculature. Through caspase-1-dependent release of proinflammatory mediators, NLRP3 drives localised immune activation within vascular tissues, contributing to endothelial dysfunction and structural remodelling of the vascular wall. This mechanistic axis provides a key inflammatory framework underpinning the progression of diabetic vascular complications [137].

### 3.7. Autophagy

Under low-grade inflammation and reduced PCSK9 expression, VSMCs mitigate cellular injury by activating autophagy or entering senescence via limited proliferation, allowing adaptation to external stressors. While these processes are considered protective, experimental studies show that VSMCs undergoing autophagy or senescence exhibit distinct morphological remodelling and metabolic reprogramming [138].

Emerging evidence suggests that PCSK9i may modulate autophagy-related mechanisms during vascular inflammation. For instance, PCSK9i suppresses IL-6–induced autophagy activation in endothelial cells [139]. Additionally, the lysosomal inhibitor NH_4_Cl reverses the downregulation of PCSK9 induced by the small molecule OY3, supporting a model in which OY3 promotes autophagy-dependent PCSK9 degradation. In contrast, the proteasome inhibitor carfilzomib exerts no significant effect, reinforcing that the degradation process bypasses the ubiquitin–proteasome system and is primarily autophagy-mediated [140]. Although current data support the involvement of PCSK9 in cardiovascular regulation via autophagic pathways, further studies are required to clarify whether it promotes or inhibits autophagy to confer cardioprotection.

### 3.8. Ferroptosis

Ferroptosis is a distinct form of programmed cell death characterized by intracellular iron accumulation, reactive oxygen species (ROS) generation via the Fenton reaction, and lipid peroxidation, ultimately leading to membrane disruption and cell death [141]. In DCM, hyperglycemia has been reported to trigger a DNA damage response that activates the DNA-dependent protein kinase complex, which may promote ferroptosis in endothelial cells [142].

Preclinical studies suggest that PCSK9 could be involved in regulating ferroptosis in cardiomyocytes, potentially through effects on mitochondrial dynamics and oxidative stress, thereby influencing disease progression. However, these findings are derived primarily from experimental models, and the clinical significance remains to be established.

Furthermore, it may represent a mechanistic link between inflammation and ferroptosis. Experimental silencing of TLR4 by siRNA has been shown to attenuate ferroptosis and ameliorate cardiac pathology in DCM models. In vitro and in vivo evidence indicates that PCSK9 expression correlates positively with TLR4 levels; overexpression of PCSK9 increases TLR4 expression in macrophages, whereas pharmacological or genetic inhibition of PCSK9 downregulates TLR4 in vascular tissues [143]. Concerning potential therapeutic strategies, agents such as Nicorandil have been reported to induce Parkin-dependent mitophagy through mitochondrial AMPKα1 activation, while concurrently inhibiting mitochondrial translocation of ACSL4, thereby attenuating the ferroptotic cascade. These findings offer a promising therapeutic strategy for targeting ferroptosis in DCM [144].

## 4. PCSK9 Involvement in the Pathological Processes of DCM

### 4.1. Cardiomyocytes

Despite its relatively low expression in cardiomyocytes, PCSK9 plays an essential role in cardiac metabolic regulation. Cardiomyocytes endogenously synthesise and secrete PCSK9, contributing to functional impairment via autocrine mechanisms [145]. PCSK9 deficiency is strongly linked to the development of cardiomyopathy, characterised by impaired mitochondrial oxidative capacity, altered substrate utilisation, and deteriorated contractile function [115]. In particular, disruption of PCSK9 affects cardiac lipid homeostasis through LDLR-independent pathways, facilitating the onset of HFpEF [146].

Experimental deletion of Pcsk9 specifically in cardiomyocytes induces a progressive, dilated cardiomyopathy-like phenotype in mice, marked by myocardial hypertrophy, fibrotic remodelling, and loss of systolic function, ultimately progressing to end-stage heart failure and early mortality [115]. Mechanistically, therapeutic silencing of the mitochondrial cholesterol transporter TSPO in CM-Pcsk9^−/−^ mice prevents cholesterol accumulation within mitochondria and restores cardiac performance, suggesting a metabolic rescue strategy in PCSK9-deficient cardiomyopathy [147].

PCSK9 contributes to regulating glucose metabolism reprogramming in cardiomyocytes under pathological stress. In DCM, elevated β-hydroxybutyric acid (βOHB) levels promote epigenetic activation of the lipocalin-2 (Lcn2) promoter via H3K9bhb modification, leading to nuclear translocation of the NF-κB/RPS3 complex and initiating proinflammatory and profibrotic gene expression programs [148]. In a murine myocardial infarction model, increased expression of Foxk1 and Foxk2 has been shown to enhance cardiomyocyte proliferation, reduce scar formation, and improve cardiac function [149]. Furthermore, CTRP9-induced upregulation of CD36 occurs independently of LRP1 and LDLR pathways, indicating a distinct regulatory mechanism. Nevertheless, PCSK9 modulates receptor-mediated signalling in cardiomyocytes, particularly its influence on LRP1-dependent pathways [77].

Evidence indicates that PCSK9i mitigates cardiac fibrosis following myocardial infarction through multiple converging mechanisms. Evolocumab improves myocardial remodelling and cardiac function in metabolic syndrome models by suppressing PCSK9-mediated activation of the NLRP3 inflammasome and its downstream Caspase-1/IL-1β axis [150]. Independently, PCSK9 modulates post-infarction fibroblast phenotypic transition via the JAK2/STAT3 pathway, further contributing to fibrotic remodelling [151].

In reperfusion injury models, PCSK9i attenuate myocardial fibrosis by inhibiting the TGF-β1/Smad3 signalling cascade and dampening local inflammatory responses [152]. Additionally, under hypoxic conditions, PCSK9 inhibition reduces collagen synthesis, fibroblast migration, and the transdifferentiation of CFs into activated myofibroblasts, with these effects mediated via Notch1/Hes1 signalling, leading to improved cardiac function [153]. Moreover, GSK-3β–dependent activation of the NLRP3 inflammasome in CFs, followed by IL-1β secretion, promotes apoptosis and pyroptosis in both CFs and cardiomyocytes (CMs), reinforcing the role of inflammasome signalling in fibrotic and inflammatory myocardial injury [154].

### 4.2. Endothelial Cells

Endothelial cells constitute a dynamic interface for vascular immune responses. Deletion of endothelial epsins delays dysfunction by inhibiting FGFR1/TGF-β–driven EndMT, thereby preserving barrier integrity under stress conditions [155,156]. Loss of the transcription factor Gata6 downregulates Cmpk2, an upstream regulator of mitochondrial DNA signalling, ultimately suppressing NLRP3 inflammasome assembly, monocyte chemotaxis, and M1-type macrophage polarisation [157]. Furthermore, S1PR2 activation induces mitochondrial fission through Rho/ROCK1-mediated phosphorylation of DRP1, which initiates NLRP3-mediated pyroptosis and exacerbates reperfusion-associated endothelial injury [136].

PCSK9 emerges as a key modulator of endothelial integrity in diabetes. Hyperglycemic and hypoxic stimuli upregulate PCSK9 expression in endothelial cells, triggering focal cell death and functional decline [127]. In diabetic conditions, PCSK9 facilitates VEGFR2 ubiquitination by enhancing its interaction with the E3 ligase NEDD4, thus suppressing VEGFR2/AKT/eNOS-ERK signalling, a crucial pathway for angiogenic repair [158]. Additionally, PCSK9 accelerates ox-LDL–induced cytotoxicity via the UQCRC1/mitochondrial ROS axis, aggravating oxidative endothelial injury [159]. Genomic analyses also identify endothelial gene variants that predict individual responsiveness to lipid-lowering strategies, offering potential for precision medicine in cardiovascular risk reduction [160].

### 4.3. Monocyte

Monocytes are key cellular mediators in the inflammatory and immunological disturbances associated with diabetes. Under hyperglycemic stress, monocytes secrete pro-cathepsin D, which facilitates blood–brain barrier transcytosis and promotes neurovascular injury [161]. Additionally, CCL2 signalling within monocytes may increase susceptibility to asthma in diabetic individuals [162]. Hyperglycemia also fosters an immunosuppressive microenvironment, notably in colorectal cancer liver metastases, by recruiting circulating monocytes via the CCL3–CCR1 axis [163].

PCSK9 exerts immunoregulatory effects on monocytes, extending beyond lipid metabolism. Neutralising PCSK9 with monoclonal antibodies attenuates CCR2-mediated chemotaxis and dampens pro-inflammatory monocyte activation in familial hypercholesterolemia [164]. In patients with stable coronary artery disease (CAD), circulating PCSK9 levels exhibit subtype-specific correlations: classical monocytes (CMs) show a positive association (R = 0.29; *p* = 0.04), while non-classical monocytes (NCMs) are inversely related (R = −0.33; *p* = 0.02) [165]. Furthermore, PCSK9–Lp(a) complex concentrations and body mass index correlate with total monocyte counts, reinforcing the lipid–immune interface [166].

Evolocumab therapy has downregulated monocyte activation markers in high-risk ASCVD populations [167]. Interestingly, PCSK9 levels correlate with HMGB1, TLR4, and NCM proportion in CAD patients not on statin therapy. However, these associations are reversed in statin-treated patients, underscoring drug-specific immunometabolic shifts. Multivariate regression confirms HMGB1, NCM%, and IM% as independent positive predictors of PCSK9 in statin-naïve individuals, and as negative predictors under statin exposure [168].

### 4.4. Macrophage

Macrophages are increasingly recognised as central mediators of inflammatory remodelling in diabetic left ventricular fibrosis and myocardial dysfunction [169]. In the context of diabetic wound pathology, lysosomal destabilisation appears to suppress NLRP3 inflammasome activation, thereby facilitating reparative responses and accelerating tissue repair [170]. Preclinical data highlight a paracrine inflammatory axis wherein keratinocytes upregulate NLRP3 expression in infiltrating macrophages via IL-1R–dependent activation of the histone demethylase JMJD3 [171].

Phenylpyruvic acid, a microbial-derived metabolite, enters macrophages via CD36-dependent uptake, binds to palmitoyl-protein thioesterase 1 (PPT1), and inhibits its depalmitoylase activity. It enhances NLRP3 palmitoylation and activates the inflammasome, leading to proinflammatory cytokine secretion and M1-like macrophage polarisation [172]. In contrast, natural compounds such as bridging sterols can attenuate inflammasome activation and exert protective anti-inflammatory effects [173]. Additionally, macrophage-specific epsin deficiency suppresses foam cell formation through altered interactions with LRP1, suggesting an anti-atherogenic role [155].

PCSK9 emerges as a potent regulator of macrophage polarisation and function. By downregulating CD36 and inhibiting lipid uptake, PCSK9 reinforces an inflammatory macrophage phenotype characterised by increased IFN-γ production and T cell stimulation [70]. LRP5–PCSK9 interaction governs intracellular lipid processing, and LRP5 facilitates PCSK9 release [125]. PCSK9 depletion significantly impairs macrophage infiltration and reduces cytokine gene expression in vascular grafts [174].

Functionally, PCSK9 operates both intracellularly and as a circulating mediator to regulate macrophage antiviral immunity and vascular inflammation [175]. Circulating PCSK9 induces macrophage activation and vein graft lesion development via LDLR-independent mechanisms [176]. Notably, PCSK9-induced NF-κB activation in macrophages augments H/R-triggered cardiomyocyte injury through heightened secretion of IL-1β and TNF-α [177].

### 4.5. Vascular Smooth Muscle Cells

PCSK9 maintains mitochondrial homeostasis in VSMCs. Its upregulation leads to enhanced phosphorylation of p38 MAPK, which subsequently activates the mitochondrial fission mediator dynamin-related protein 1 (DRP1), resulting in marked changes in mitochondrial morphology and function. It is accompanied by increased intracellular reactive oxygen species (ROS) levels, ultimately impairing cellular function and viability. DRP1, as a central regulator of mitochondrial dynamics, appears to be a critical downstream effector of PCSK9, suggesting the presence of a feedback regulatory loop between PCSK9 and DRP1 [178]. Additionally, under diabetic conditions, oxidative stress activates the AMPK–Cx43–NLRP3 axis, contributing to extracellular matrix remodelling in gastric smooth muscle, as observed in diabetic gastroparesis models [179].

PCSK9 also promotes vascular ageing through multiple mechanisms. Its overexpression downregulates apolipoprotein E receptor 2, promotes polyploidization of VSMCs, and accelerates senescence-associated vascular changes [138]. Conversely, PCSK9 deficiency alleviates atherosclerosis-associated ageing phenotypes by significantly reducing neointimal thickening, vascular stenosis, and collagen deposition [174]. In diabetes, the metabolite palmitic acid also induces VSMCs’ senescence by activating the Dll4 signalling pathway in macrophages. This process depends on activating the TLR4/ERK/FOXC2 pathway, which increases plaque instability and vulnerability [180].

## 5. Research Progress on PCSK9 in Other Fields

### 5.1. PCSK9 and Ischemia/Reperfusion Injury

PCSK9 is increasingly recognised as a pathophysiological mediator and therapeutic target in ischemia–reperfusion (I/R) injury following ST-segment elevation myocardial infarction (STEMI). Experimental models indicate that PCSK9 inhibition via evolocumab protects cardiomyocytes by suppressing LIAS-dependent cuproptosis, a copper-induced cell death pathway implicated in I/R-associated myocardial damage [10].

Clinically, serial measurements in STEMI patients reveal a delayed but sustained rise in circulating PCSK9 levels within 48 h post-percutaneous coronary intervention. While early elevations (at 24 h) do not correlate with myocardial or microvascular injury, higher PCSK9 levels at 48 h are strongly associated with intramyocardial haemorrhage, microvascular obstruction, infarct expansion, and adverse clinical prognosis [9]. These findings highlight the temporal relevance of PCSK9 dynamics and reinforce its potential as a biomarker and therapeutic target in acute myocardial infarction management.

### 5.2. PCSK9 and Cancer

PCSK9 has been proposed as a multifunctional regulator of tumour progression and metastasis through its ability to reprogram the tumour microenvironment and intracellular signalling networks. It modulates metastatic competency by downregulating low-density lipoprotein receptor–related protein 1 (LRP1), thereby derepressing transcriptional programs associated with invasion and dissemination [181]. Beyond LRP1 suppression, experimental data indicate that PCSK9 could contribute to cancer progression by promoting epithelial–mesenchymal transition (EMT), activating the PI3K/AKT signalling pathway, and influencing macrophage polarisation toward pro-inflammatory and pro-metastatic phenotypes [182].

Mechanistically, PCSK9 overexpression has been associated with enhanced phosphorylation of mitogen-activated protein kinases (MAPKs), including p38MAPK, ERK1/2, and JNK, which may facilitate tumour cell migration and invasion. Gene silencing of PCSK9 in model systems attenuates this phosphorylation cascade and impairs tumour cell motility [183]. In anaplastic thyroid carcinoma, PCSK9 drives the lysosome-dependent degradation of E-cadherin, thereby facilitating EMT and tumour spread. Loss-of-function mutations in p53—especially the R248Q variant—further upregulate PCSK9 transcription by disrupting its repression, amplifying tumour malignancy. Targeted inhibition using the small molecule PF-846 suppresses PCSK9 expression and significantly reduces tumour proliferation and metastasis in vitro and in vivo [184].

Moreover, PCSK9 may also influence the metabolic adaptation of cancer cells in distinct microenvironments. In pancreatic cancer, low PCSK9 expression promotes extracellular cholesterol uptake, leveraging hepatic cholesterol abundance. Conversely, elevated PCSK9 levels shift cancer cells toward intrinsic cholesterol biosynthesis and antioxidant production, enhancing their survival in oxidative niches such as the lungs [185]. These observations highlight PCSK9 as a potential modulator of tumour biology. However, current evidence is derived mainly from in vitro and animal studies, and the translational and clinical relevance remains to be determined.

## 6. Conclusions and Perspectives

PCSK9 has garnered recognition beyond its canonical role in lipid metabolism, emerging as a pleiotropic effector in DCM. In this review, we delineate the multifaceted roles of PCSK9 in metabolic dysregulation, immune activation, and programmed cell death, and examine its cell-specific effects in cardiomyocytes, endothelial cells, macrophages, monocytes, and vascular smooth muscle cells. Mechanistically, PCSK9 contributes to chronic lipotoxic and inflammatory signalling via CD36- and TLR4-dependent pathways, fostering maladaptive cardiac remodelling in the diabetic milieu.

Pharmacologically, PCSK9 inhibition offers dual benefits—practical LDL-C reduction and favourable metabolic neutrality. Evidence from clinical trials and Mendelian randomisation studies suggests that PCSK9i do not increase the incidence of NODM, and that associated T2DM risk from partial loss-of-function mutations is minimal. Beyond lipid-lowering, PCSK9i demonstrate anti-inflammatory, anti-fibrotic, and cardio-metabolic effects that support their use in diabetic populations, especially among statin-intolerant patients or those with severe dysmetabolism.

Notable therapeutic candidates include Tafolecimab, which induces significant and sustained LDL-C reductions with favourable safety in Chinese HeFH cohorts at biweekly and monthly dosing intervals [186], and AZD0780, an orally administered PCSK9 modulator, provides potent dose-dependent LDL-C reductions and favourable pharmacokinetics, supporting further clinical development [187].

Critical knowledge gaps remain. Future studies must clarify the intracellular pathways modulated by PCSK9 in DCM, including mitochondrial bioenergetics, ferroptosis, and autophagy. Additionally, the pathophysiologic relevance of PCSK9 across various diabetic subtypes (T1DM, T2DM, GDM) must be elucidated through multicenter, phenotype-specific cohorts. Finally, combinatorial regimens with GLP-1 receptor agonists or SGLT2 inhibitors require rigorous investigation to determine additive or antagonistic effects. Of note, targeted epigenetic editing of the PCSK9 promoter has emerged as a promising strategy for personalised intervention.

In summary, targeting PCSK9 represents a novel and promising strategy for DCM management, potentially improving both survival outcomes and quality of life in patients with diabetes-related heart disease.

## Figures and Tables

**Figure 1 biomolecules-15-01240-f001:**
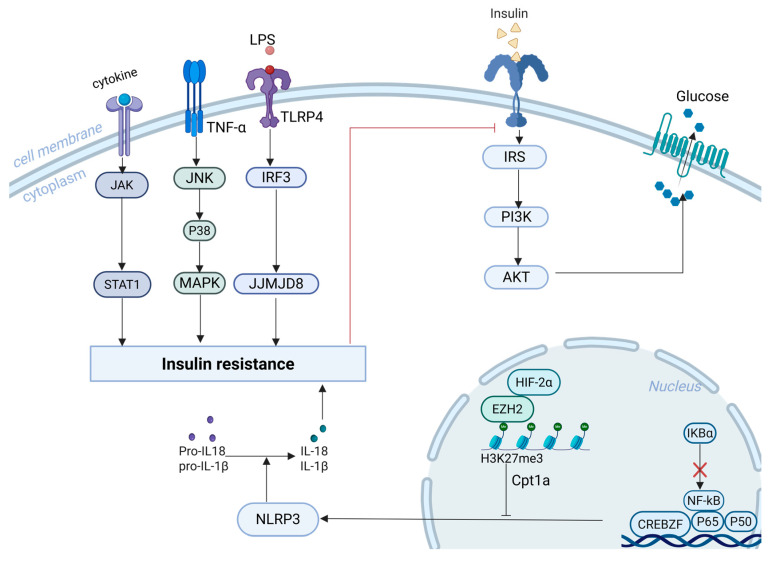
Mechanistic Pathways Linking Inflammation and Insulin Resistance. Legend: the red lines represent inhibitory effects.

## Data Availability

No new data were created or analyzed in this study.

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
