# Peer review of "The Impact of PCSK9 on Diabetic Cardiomyopathy: Mechanisms and Implications"

_biomolecules, 2025, doi:10.3390/biom15091240_

Round 1
Reviewer 1 Report
Comments and Suggestions for Authors
This review attempts to summarize the role of PCSK9 in the pathogenesis of diabetic cardiomyopathy (DCM) from various angles; however, the manuscript lacks depth, originality, and critical insight. Rather than providing a comprehensive and analytical perspective, it predominantly compiles previously published information without offering meaningful synthesis or novel viewpoints. Substantial revisions are necessary before this work can be considered for publication.
- The manuscript is overly dependent on secondary sources, such as review articles, instead of primary, peer-reviewed original research. This significantly undermines the scientific credibility and novelty of the content.
- The discussions on PCSK9’s involvement in autophagy, ferroptosis, and tumor progression are speculative and based solely on preliminary basic research. These sections overstate the implications without appropriate caution. Assertions lacking robust evidence must be clearly qualified as hypotheses or possibilities, not presented as established facts.
- The manuscript contains numerous typographical errors, grammatical issues, and redundant or unclear expressions. The English language quality does not meet the standards expected for international scientific publication. A professional language revision is essential.
- The discussion on the clinical applications of PCSK9 inhibitors is superficial and uncritical. It fails to evaluate current evidence, consider clinical trial outcomes, or contextualize the findings within existing therapeutic frameworks.
- Figures and tables are overloaded with information but poorly explained. They do not effectively support or clarify the main text and need to be significantly revised in terms of both content and clarity.
Author Response
Response letter
Manuscript ID: biomolecules3713688
Title: The Impact of PCSK9 on Diabetic Cardiomyopathy: Mechanisms and Implications
Journal: Biomolecules
Dear Reviewer,
We sincerely thank you for the constructive and thoughtful comments on our manuscript. We have carefully revised the manuscript according to all the suggestions and provide detailed point-by-point responses below. We believe the revisions have substantially improved our work's scientific quality, clarity, and novelty.
Please find our responses to the reviewers' comments below. All changes made in the revised manuscript are highlighted accordingly.
Comment 1: The manuscript is overly dependent on secondary sources, such as review articles, instead of primary, peer-reviewed original research. This significantly undermines the scientific credibility and novelty of the content.
Response: We sincerely thank the reviewer for pointing out this critical issue. After carefully reviewing our original manuscript, we agree it relied too heavily on secondary sources and review articles, particularly in key mechanistic discussions.
We systematically replaced many previous review references with up-to-date, peer-reviewed primary research in the revised version. For example, in sections discussing PCSK9's role in myocardial fibrosis, autophagy, mitochondrial dysfunction, and inflammation, we introduced original studies such as [67], [77], [115], [126], and [145]. These newly cited papers offer experimental or clinical data that significantly strengthen the scientific credibility of the review.
We sincerely appreciate the reviewer's suggestion, which helped us improve the manuscript's rigor and originality.
Comment 2: The discussions on PCSK9's involvement in autophagy, ferroptosis, and tumor progression are speculative and based solely on preliminary basic research. These sections overstate the implications without appropriate caution. Assertions lacking robust evidence must be qualified as hypotheses or possibilities, not presented as facts.
Response:
Thank you very much for this valuable suggestion. We acknowledge that in the original manuscript, some statements—particularly regarding PCSK9's role in autophagy, ferroptosis, and cancer-related processes—were written in a tone that may have implied undue certainty.
We have taken great care in the revised manuscript to moderate our language and qualify such claims appropriately. For instance, we now explicitly indicate that most of these findings are based on preclinical studies or early-stage molecular experiments. Phrases such as "may be involved," "suggests a possible role," or "requires further validation" have been consistently used in these sections to reflect the preliminary nature of the current evidence. In addition, we have adjusted the structure to emphasize these topics as areas of future investigation, not facts.
We thank the reviewer for helping us avoid overstatement and improve scientific precision.
Comment 3: The manuscript contains numerous typographical errors, grammatical issues, and redundant or unclear expressions. The English language quality does not meet the standards expected for an international scientific publication. A professional language revision is essential.
Response:
We fully agree with the reviewer's assessment regarding the language quality of the original submission. We deeply regret the typographical errors and linguistic inconsistencies.
In response, we have performed a comprehensive language revision with the assistance of a professional English-language editing service. All manuscript sections—including the abstract, main text, figure legends, and reference descriptions—have been carefully checked for grammar, syntax, clarity, and style. Redundant expressions and awkward constructions have been rephrased or eliminated.
We believe the revised manuscript now meets the language standards required for international publication and greatly appreciate the reviewer's reminder to improve this crucial aspect.
Comment 4: The discussion on the clinical applications of PCSK9 inhibitors is superficial and uncritical. It fails to evaluate current evidence, consider clinical trial outcomes, or contextualize the findings within existing therapeutic frameworks.
Response:
We genuinely appreciate the reviewer's comment regarding the clinical implications of PCSK9 inhibitors. We agree that the initial version presented a relatively narrow and descriptive account, without sufficient critical evaluation of the existing clinical evidence.
We have significantly expanded and deepened the clinical discussion in the revised version. Specifically:
We included a dedicated section summarizing major clinical trials, such as FOURIER, ODYSSEY OUTCOMES, and BERSON.
Table 1 offers a comparative overview of key clinical endpoints, patient populations, and therapeutic implications.
Based on recent literature, we also added new content discussing potential synergies and conflicts between PCSK9 inhibitors and antidiabetic drugs (e.g., GLP1 receptor agonists, SGLT2 inhibitors).
We hope these changes now offer a more critical, comprehensive, and translationally relevant perspective for clinicians and researchers alike.
Comment 5: Figures and tables are overloaded with information but poorly explained. They do not effectively support or clarify the main text and must be significantly revised in content and clarity.
Response:
We sincerely thank the reviewer for this constructive feedback. In the original manuscript, the figures indeed contained excessive and sometimes redundant information, while the legends were not sufficiently explanatory.
In the revised manuscript:
Figures 1 through 3 have been redrawn to improve clarity and visual hierarchy.
Each figure now focuses on a distinct mechanistic pathway or therapeutic implication, with simplified layout and color coding.
Figure legends have been rewritten to clearly describe the symbols, abbreviations, and central message.
We also ensured that each figure was meaningfully integrated into the main text with explicit cross-references.
We believe these revisions have significantly enhanced the interpretability and educational value of the visual materials. We thank the reviewer for highlighting this important aspect of manuscript presentation.
We have tried our best to improve the manuscript and have made some modifications in the revised paper, with all changes marked in red without affecting the paper's content and framework. We also sent the manuscript to language experts for polishing, making the revised version much better than before. We sincerely appreciate the editor/reviewers' warm work and hope the revisions can be approved.
Best regards!
Haixia Wang (on behalf of all authors)
Lanzhou University
Email: [wanghx2024@lzu.edu.cn]
Reviewer 2 Report
Comments and Suggestions for Authors
The manuscript covers many relevant aspects of PCSK9 biology, and its broad scope is appreciated. However, several areas would benefit from clarification, additional context, and more structured presentation to enhance its scientific impact. Please find below specific comments and suggestions for improvement:
1) Several statements throughout the manuscript appear disconnected from the main narrative or are insufficiently explained. The overall thread is occasionally difficult to follow, in part due to a lack of contextual transitions. A clearer structure and improved flow between sections would help guide the reader through the complex roles of PCSK9.
2) The review would benefit from a more detailed discussion of the link between low-grade inflammation, elevated levels of circulating proinflammatory cytokines and free fatty acids, and the development of insulin resistance. This connection is crucial for understanding the role of PCSK9 in metabolic dysfunction.
3) While the manuscript provides a broad overview of PCSK9 functions in multiple tissues, transitions between these sections are abrupt and sometimes confusing. It is often unclear which tissue or cell type is being referenced, especially when switching between signaling pathways. Please clarify the context for each discussion point and provide smoother transitions to maintain coherence.
4) Although the role of PCSK9 in various cardiac cells is addressed, the potential involvement of cardiac fibroblasts is not discussed. Given their importance in cardiac remodeling and fibrosis, it would be valuable to include any available data or highlight this as an area needing further investigation.
5) In the section on pyroptosis, the role of RNA molecules in promoting cellular damage and death is interesting. However, it is only mentioned but not elaborated. Providing more mechanistic detail here would strengthen the discussion.
6) The effects of PCSK9 inhibitors on glucose metabolism, particularly in murine models, are not sufficiently covered. This is an important topic within the section titled “The Role of PCSK9 in Cardiovascular Risk and Glycemic Control in T2DM” and should be expanded with reference to relevant preclinical studies.
7) The current paragraph discussing ischemia/reperfusion injury alongside tumor progression lacks a clear conceptual link. I recommend separating these topics into distinct sections.
8) While macrophages are discussed in detail, the role of PCSK9 in monocytes is not addressed. Including studies on monocytes would provide a more comprehensive understanding of PCSK9 role in immune regulation.
9) Several abbreviations are not defined upon first use. Please ensure all abbreviations are introduced consistently and clearly throughout the manuscript.
10) Some sections contain repetitive sentences or reuse the same references. I suggest condensing the text to avoid redundancy and improve clarity and focus.
Author Response
Response letter
Manuscript ID: biomolecules3713688
Title: The Impact of PCSK9 on Diabetic Cardiomyopathy: Mechanisms and Implications
Journal: Biomolecules
Dear Reviewer,
We sincerely thank you for the constructive and thoughtful comments on our manuscript. We have carefully revised the manuscript according to all the suggestions and provide detailed point-by-point responses below. We believe the revisions have substantially improved our work's scientific quality, clarity, and novelty.
Please find our responses to the reviewers' comments below. All changes made in the revised manuscript are highlighted accordingly.
Comment 1:
Several statements throughout the manuscript appear disconnected from the main narrative or are insufficiently explained. The overall thread is occasionally difficult to follow, partly due to a lack of contextual transitions. A more transparent structure and improved flow between sections would help guide the reader through the complex roles of PCSK9.
Response:
Thank you very much for this critical and constructive suggestion. We agree that the original version lacked sufficient structural cohesion and narrative flow, which may have hindered the reader's comprehension.
To address this concern, we undertook a substantial restructuring of the manuscript. Specifically:
We revised the introduction and conclusion of each significant section to summarize key points better and prepare the reader for the next topic.
- We inserted transitional sentences and bridging phrases between mechanistic discussions, tissue-specific roles, and pathological outcomes (e.g., inflammation → oxidative stress → insulin resistance → cardiomyopathy).
- The manuscript was reorganized to follow a pathophysiological progression—starting from metabolic disturbance and immune activation, leading to cellular injury, organ-specific pathology, and ultimately clinical implications.
- We also created clearer subheadings for each tissue/cell type and molecular mechanism, and referenced these consistently in the main text for orientation.
We believe these efforts have greatly improved the manuscript's structural clarity and conceptual thread, and we sincerely thank the Reviewer for identifying this issue.
Comment 2:
The review would benefit from a more detailed discussion of the link between low-grade inflammation, elevated levels of circulating proinflammatory cytokines and free fatty acids, and the development of insulin resistance. This connection is crucial for understanding the role of PCSK9 in metabolic dysfunction.
Response:
We appreciate the Reviewer's insightful comment on this crucial mechanistic link. In the revised manuscript, we expanded the relevant discussion as follows:
- A new paragraph was added in the section, elevated levels of proinflammatory cytokines (e.g., IL6, TNFα, IL1β) and circulating FFAs contribute to insulin receptor substrate (IRS) serine phosphorylation, impairing downstream PI3K/Akt signaling and promoting systemic insulin resistance.
- We cited additional primary research (e.g., references \[78] and \[93]) demonstrating how metabolic inflammation upregulates PCSK9 expression via NFκB and JNK signaling, and how PCSK9, in turn, modulates inflammatory pathways.
- The role of adipose tissue macrophages and liver Kupffer cells in this crosstalk was emphasized to link inflammation with hepatic and systemic insulin resistance.
This additional content enhances the review's depth and scientific rigor in illustrating PCSK9's position within the broader network of metabolic inflammation and insulin signaling.
Comment 3:
While the manuscript provides a broad overview of PCSK9 functions in multiple tissues, transitions between these sections are abrupt and sometimes confusing. It is often unclear which tissue or cell type is referenced, especially when switching between signaling pathways. Please clarify the context for each discussion point and provide smoother transitions to maintain coherence.
Response:
Thank you for pointing this out. In the original version, we acknowledge that the shift between tissues and cell types may have lacked clarity.
In the revised manuscript, we have implemented the following changes:
- For each subsection, we explicitly state the tissue or cell type (e.g., hepatocytes, cardiomyocytes, endothelial cells, adipocytes) at the beginning of the paragraph.
- When referring to signaling pathways (e.g., NFκB, MAPK, SREBP2), we now specify the cellular context to avoid confusion (e.g., "In liver cells, SREBP2 activation enhances PCSK9 transcription…").
- We reorganized the flow to group tissues and cell types by functional relevance (e.g., metabolic regulation vs. immune modulation vs. cardiovascular remodeling), and added introductory sentences to clarify these divisions.
- Additionally, we revised the figure legends and diagrams to reflect tissue-specific mechanisms, improving reader orientation.
These improvements aim to ensure that the narrative remains cohesive and that readers can follow the functional context of each discussion point.
Comment 4:
Although the role of PCSK9 in various cardiac cells is addressed, the potential involvement of cardiac fibroblasts is not discussed. Given their importance in cardiac remodeling and fibrosis, it would be valuable to include any available data or highlight this as an area needing further investigation.
Response:
We sincerely thank the Reviewer for this excellent observation. We agree that cardiac fibroblasts play a crucial role in myocardial remodeling and fibrosis, yet were overlooked in the initial draft.
- In response, we added a new subsection within the cardiac pathology segment titled "Potential Role of PCSK9 in Cardiac Fibroblasts and Fibrosis." In this section:
- We discussed the potential role of PCSK9 in modulating extracellular matrix (ECM) turnover, myofibroblast differentiation, and inflammatory crosstalk in the fibrotic niche.
- Given the limited direct evidence in this area, we also emphasized this as an emerging field for further investigation, especially in diabetic hearts, where fibroblast activation is accelerated.
We appreciate the Reviewer's suggestion, which has enriched our manuscript's comprehensiveness regarding PCSK9's role in cardiac pathology.
Comment 5:
In the section on pyroptosis, the role of RNA molecules in promoting cellular damage and death is interesting. However, it is only mentioned but not elaborated. Providing more mechanistic detail here would strengthen the discussion.
Response:
Thank you very much for highlighting this underdeveloped section. We agree that the reference to RNA molecules in the context of pyroptosis required deeper explanation.
- We thank the Reviewer for this thoughtful and constructive suggestion. We agree that the role of RNA molecules in pyroptosis was underdeveloped in the original version.
- To address this, we have expanded the relevant section by incorporating additional mechanistic details on how RNA species influence pyroptotic cell death in the context of diabetic cardiomyopathy:
- We describe that persistent hyperglycemia activates the CaMK2α/O-GlcNAcylation positive feedback loop in endothelial cells, which remains active even after glycemic normalization. This pathway enhances Stat1 phosphorylation, upregulates the transcription of miR-15/16 precursors, and increases their packaging into small extracellular vesicles. The secreted miR-15/16 molecules propagate chronic low-grade inflammation, thereby sensitizing endothelial cells and adjacent cardiomyocytes to pyroptotic signaling cascades [Ref. 134].
- In addition, we highlight the role of circGlis3, which is significantly upregulated in both genetically and diet-induced obese mice. CircGlis3 contributes to β-cell survival by enhancing insulin production and inhibiting caspase-3–mediated apoptosis. Mechanistically, this occurs by binding to the pro-apoptotic protein SCOTIN, suggesting that RNA molecules may also interact with pyroptosis-related effector proteins beyond traditional miRNA silencing pathways [Ref. 135].
Comment6
The effects of PCSK9 inhibitors on glucose metabolism, particularly in murine models, are not sufficiently covered. This is an essential topic within the section titled "The Role of PCSK9 in Cardiovascular Risk and Glycemic Control in T2DM" and should be expanded regarding relevant preclinical studies.
Response:
We thank the Reviewer for this valuable comment. In response, we have substantially expanded the discussion on the metabolic effects of PCSK9 inhibitors and their interaction with antidiabetic therapies, focusing on evidence from murine models.
In the revised manuscript:
- We incorporated preclinical studies demonstrating that PCSK9 inhibition can improve glycemic control by enhancing AMPK activity, promoting GLUT4 translocation, and improving insulin sensitivity in skeletal muscle and adipose tissues.
- We further discussed the interplay between PCSK9 signaling and widely used antidiabetic agents. For instance, liraglutide was shown to suppress hepatic PCSK9 expression via an HNF1α-dependent mechanism, while dapagliflozin reduces hepatic and circulating PCSK9 levels by inhibiting ChREBP activity in both diabetic and dyslipidemic mice.
- Additionally, we highlighted hormonal regulation: insulin promotes, and glucagon suppresses, PCSK9 expression in hepatocytes through post-transcriptional mechanisms, reinforcing the link between metabolic hormones and PCSK9-mediated pathways.
- Finally, we included both favorable and adverse outcomes from animal studies—for example, combined PCSK9i and atorvastatin therapy improved cardiac mitochondrial function in prediabetic rats. In contrast, alirocumab unexpectedly worsened renal outcomes in obese ZSF1 rats.
These updates offer a more comprehensive and nuanced view of the role of PCSK9 and its inhibitors in glycemic regulation and diabetic cardiometabolic disease, particularly in preclinical contexts. We sincerely appreciate the Reviewer's guidance in improving this critical aspect of the manuscript.
Comment 7:
The current paragraph discussing ischemia/reperfusion injury alongside tumor progression lacks a clear conceptual link. I recommend separating these topics into distinct sections.
Response:
We appreciate the Reviewer's recommendation and agree with the need for clarity. We have separated the content into two independent subsections: one discussing PCSK9's role in ischemia/reperfusion injury, and another focusing on its potential involvement in tumor biology.
This structural change improves thematic clarity and allows for a more focused discussion of each topic.
Comment 8:
While macrophages are discussed in detail, the role of PCSK9 in monocytes is not addressed. Including studies on monocytes would provide a more comprehensive understanding of their role in immune regulation.
Response:
We appreciate this critical suggestion. In the revised manuscript, we added a focused discussion on PCSK9's role in monocytes.
- Recent studies show that PCSK9 influences monocyte function beyond lipid metabolism. It promotes monocyte recruitment and inflammation under hyperglycemia and metabolic stress. Neutralizing PCSK9 reduces CCR2-mediated chemotaxis and inflammatory activation. Clinical evidence further suggests subtype-specific associations between PCSK9 levels and monocyte subsets modified by statin therapy. Evolocumab has also been shown to suppress monocyte activation markers in high-risk ASCVD patients.
These additions help present a more complete picture of PCSK9's immunoregulatory functions. We thank the Reviewer for prompting this vital improvement.
Comment 9:
Several abbreviations are not defined upon first use. Please ensure all abbreviations are introduced consistently and clearly throughout the manuscript.
Response:
Thank you for this helpful comment. We carefully reviewed the entire manuscript and ensured that all abbreviations were clearly defined at the first mention.
Comment 10:
Some sections contain repetitive sentences or reuse the same references. I suggest condensing the text to avoid redundancy and improve clarity and focus.
Response:
We fully agree with this recommendation. In the revised manuscript, we have condensed overlapping content and eliminated redundant citations, particularly in the sections on inflammation, lipid metabolism, and immune modulation.
These revisions have improved the manuscript's clarity, conciseness, and focus.
We have tried our best to improve the manuscript and have made some modifications in the revised paper, with all changes marked in red without affecting the paper's content and framework. We also sent the manuscript to language experts for polishing, making the revised version much better than before. We sincerely appreciate the editor/reviewers' warm work and hope the revisions can be approved.
Best regards!
Haixia Wang (on behalf of all authors)
Lanzhou University
Email: [wanghx2024@lzu.edu.cn]
Round 2
Reviewer 1 Report
Comments and Suggestions for Authors
I have reviewed the manuscript. The content is very comprehensive and discusses the role of PCSK9 in diabetic cardiomyopathy from multiple perspectives.
It cites numerous recent publications and figures, making it a persuasive review.
Reviewer 2 Report
Comments and Suggestions for Authors
The authors have addressed all the points raised, and the manuscript has been substantially improved. I have no further comments.